# A MULTIOBJECTIVE CONTINUATION METHOD TO COMPUTE THE REGULARIZATION PATH OF DEEP NEURAL NETWORKS

## ABSTRACT

Sparsity is a highly desired feature in deep neural networks (DNNs) since it ensures numerical efficiency, improves the interpretability of models (due to the smaller number of relevant features), and robustness. For linear models, it is well known that there exists a *regularization path* connecting the sparsest solution in terms of the $\ell^1$ norm (i.e., zero weights) and the non-regularized solution. Very recently, there was a first attempt to extend the concept of regularization paths to DNNs by means of treating the empirical loss and sparsity ($\ell^1$ norm) as two conflicting criteria and solving the resulting multiobjective optimization problem. However, due to the non-smoothness of the $\ell^1$ norm and the high number of parameters, this approach is not very efficient from a computational perspective. To overcome this limitation, we present an algorithm that allows for the approximation of the entire Pareto front for the above-mentioned objectives in a very efficient manner. We present numerical examples using both deterministic and stochastic gradients. We furthermore demonstrate that knowledge of the regularization path allows for a well-generalizing network parametrization. To the best of our knowledge, this is the first algorithm to compute the regularization path for non-convex problems with millions of degrees of freedom.

## 1 INTRODUCTION

Machine Learning (ML) and in particular deep neural networks (DNNs) are nowadays an integral part of numerous applications such as data classification, image recognition, time series prediction, and language processing. Their importance continues to grow at great speed across numerous disciplines and applications, and the increase in available computational capacity allows for the construction of larger models. However, these advances also increase the challenges regarding the construction and training of DNNs, e.g., the required training data, the training efficiency, and the adaptability to changing external factors. This leads to the task of simultaneously fulfilling numerous, sometimes contradictory goals in the best possible way.

Multiobjective optimization addresses the problem of optimizing several conflicting objectives. The issue of having to trade between multiple, conflicting criteria is a universal problem, such as the need to have an optimal tradeoff between cost and quality in a production process. In a similar manner, conflicting criteria occur in various ways in ML. The main task is thus to identify the set of optimal trade-off solutions (the *Pareto set*) between these conflicting criteria. This concerns multitask problems (Sener & Koltun, 2018), but also the training itself, where we want to minimize the training loss, obtain sparse models and improve generalization.

Interestingly, we found that many papers on multicriteria machine learning do not address the true nature of multiobjective optimization. The reason for this, is that when choosing very large neural network architectures or even considering task-specific layers (Sener and Koltun, 2018), the different tasks are no longer conflicting. The network is simply too powerful such that both tasks can be optimally handled simultaneously and there is no tradeoff. From an optimization point of view, the Pareto front collapses into a single point. However, considering the strongly growing carbon footprint of AI Gibney (2022), there is a growing interest in smaller models that are better tailored to specific tasks. This is why we propose the use of models that are smaller and adapted to a

certain task. While this will reduce the general applicability, multicriteria optimization can help us to determine a set of compromising network architectures, such that we can adapt networks to specific situations online (multicriteria decision making).

The joint consideration of loss and $\ell^1$ regularization is well-studied for linear systems. However, it is much less understood for the nonlinear problems that we face in deep learning. In DNN training, the regularization path is usually not of interest. Instead, methods aim to find a single, suitable trade-off between loss and $\ell^1$ norm (Chen et al., 2021; Bungert et al., 2022; Fu et al., 2020; 2022; Lemhadri et al., 2021). When interpreting the $\ell^1$ regularization problem as a multiobjective optimization problem (MOP), a popular approach to obtain the entire solution set (the *Pareto set*) is via *continuation methods* Hillermeier (2001); Schütze et al. (2005). They usually consist of a predictor step (along the tangent direction of the Pareto set) and a corrector step that converges to a new point on the Pareto set close by. However, as the $\ell^1$ norm is non-smooth, classical manifold continuation techniques fail. Due to this fact, a first extension of regularization paths from linear to nonlinear models was recently presented in (Bieker et al., 2022), where continuation methods were extended to non-smooth objective functions. Although this extension provides a rigorous treatment of the problem, it results in a computationally expensive algorithm, which renders it impractical for DNNs of realistic dimensions.

As research on regularization paths for nonlinear loss functions has focused on small-scale learning problems until now, this work is concerned with large-scale machine learning problems. The main contributions of this paper include:

- The extension of regularization paths from linear models to high dimensional nonlinear deep learning problems. In fact, our algorithm is the first to provide the entire Pareto front for a problem with millions of degrees of freedom.
- The demonstration of the usefulness of multiobjective continuation methods for the generalization properties of DNNs, as we can find sweet spots on the Pareto front very easily.
- A step towards more resource-efficient ML by beginning with very sparse networks and slowly increasing the number of weights. This is in complete contrast to the standard pruning approaches, where we start very large and then reduce the number of weights.

A comparison of our approach is further made with the standard approach of weighting the individual losses using an additional hyperparameter. The remainder of the paper is organized as follows. First, we discuss related works, before introducing our continuation method. We then present a detailed discussion of our extensive numerical experiments.

## 2 RELATED WORKS

### 2.1 MULTIOBJECTIVE OPTIMIZATION

In the last decades, many approaches have been introduced for solving nonlinear (Miettinen, 1998), non-smooth (Poirion et al., 2017), non-convex (Miettinen, 1998), or stochastic (Mitrevski et al., 2021) multiobjective optimization problems, to name just a few special problem classes. In recent years, researchers have further begun to consider multiobjective optimization in machine learning (Qu et al., 2021; Jin & Sendhoff, 2008; Deb, 2011; Zhang & Li, 2007) and deep learning (Sener & Koltun, 2018; Ruchte & Grabocka, 2021). We provide an overview of the work that is most pertinent to our own and direct readers to the works done by (Sener & Koltun, 2018) and (Bieker et al., 2022). Different methods have been proposed to obtain Pareto optimal solutions such as *evolutionary algorithms* which use the evolutionary principles inspired by nature by evolving an entire population of solutions (Deb, 2001). *Gradient-based techniques* extend the well-known gradient techniques from single-objective optimization to multiple criteria problems (Peitz & Dellnitz, 2018). *Set Oriented Methods* provide an approximation of the Pareto set through box coverings and often suffer from the curse of dimensionality which makes applications in ML very expensive (Dellnitz et al., 2005). Another deterministic approach are *scalarization methods* which involve the transformation of MOP into (a series of) single objective optimization problems (Eichfelder, 2008). These can then be solved using standard optimization techniques. Examples include the weighted sum method, epsilon constraint, Chebyshev scalarization, goal programming, min-max and augmented epsilon constraint method (Ehrgott 2008, Kim & Weck 2006). Some drawbacks exist for the latter methods,

most notably the incapability of some to handle non-convex problems, as well as the challenge to obtain equidistant coverings. Notably, these drawbacks are most severe in the weighted sum method, which is by far the most widely applied approach in ML when considering multiple criteria at the same time (such as regularization terms). Moreover, the weighting has to be made a priori, which makes selecting useful trade-offs much harder (Ehrgott, 2008; Sener & Koltun, 2018).

*Continuation methods* are another approach for the computation of the solutions of MOPs. These methods were initially developed to solve complex problems starting from simple ones, using homotopy approaches[1]. Also, they were used in predictor-corrector form to track manifolds (Allgower & Georg, 1990; Chow et al., 1991). In the multiobjective setting, one can show that the Pareto set is a manifold as well[2], if the objectives are sufficiently regular (Hillermeier, 2001). Hence, the continuation method then becomes a technique to follow along the Pareto set in two steps; **a)** predictor step along the tangent space of the Pareto set which forms in the smooth setting a manifold of dimension *m - 1* (where $m$ is the number of objective functions) (Hillermeier, 2001) and **b)** a corrector step that obtains the next point on the Pareto set which is often achieved using multiobjective gradient descent. Schütze et al. (2005) used the predictor-corrector approach to find points that satisfy the Karush-Kuhn-Tucker (KKT) condition and to further identify other KKT points in the neighbourhood.

The continuation method has further been extended to regularization paths in machine learning (ML) for linear models (Park & Hastie, 2007; Guha et al., 2023). The regularization paths for nonlinear models as a multiobjective optimization problem have been introduced recently although limited to small dimensions since dealing with non-smoothness is difficult (Bieker et al., 2022).

## 2.2 MULTICRITERIA MACHINE LEARNING

In the context of multicriteria machine learning, several approaches have been used such as *evolutionary algorithms* (Deb et al., 2002; Bernadó i Mansilla & Garrell i Guiu, 2001; Jin & Sendhoff, 2008) or *gradient-based methods* (Sener & Koltun, 2018; Mitrevski et al., 2021). However, only a few of these methods address high-dimensional deep learning problems or attempts to compute the entire Pareto front. Furthermore, as discussed in the introduction, many researchers have introduced architectures that are so powerful that even with the inclusion of the task-specific parts, the Pareto front collapses into a single point (e.g., Sener & Koltun (2018)). In the same way, the regularization path is usually not of much interest in DNN training. Instead, most algorithms yield a single trade-off solution for loss and $\ell^1$ norm that is usually influenced by a weighting parameter (Chen et al., 2021; Bungert et al., 2022; Fu et al., 2022; Lemhadri et al., 2021). However, we here want to pursue a more sustainable path with well-distributed solutions here, and we are interested in truly conflicting criteria. The entire regularization path for DNNs (i.e., a MOP with training loss versus $\ell^1$ norm) was computed in (Bieker et al., 2022). However, even though the algorithm provably yields the Pareto front, the computation becomes intractable in very high dimensions. Hence, the need to develop an efficient method to find the entire regularization path and Pareto front for DNNs.

For high-dimensional problems, gradient-based methods have proven to be the most efficient. Examples are the steepest descent method (Fliege & Svaiter, 2000), projected gradient method (Drummond & Iusem, 2004), proximal gradient method (Tanabe et al., 2019; Chen et al., 2021) and recently accelerated proximal gradient (Tanabe et al., 2023). Previous approaches for MOPs often assume differentiability of the objective functions but the $\ell^1$ norm is not differentiable, so we use the multiobjective proximal gradient (MPG) to ensure convergence. MPG has been described by Tanabe et al. (2019) as a descent method for unconstrained MOPs where each objective function can be written as the sum of a convex and smooth function, and a proper, convex, and lower semicontinuous but not necessarily differentiable one. Simply put, MPG combines the proximal point and the steepest descent method and is a generalization of the iterative shrinkage-thresholding algorithms (ISTA) (Combettes & Wajs, 2005; Beck & Teboulle, 2009) to multiobjective optimization problems.

---

[1]These are approaches that involve starting at a simple-to-calculate solution and then continuously vary some parameter to increase the problem difficulty step by step, until finally arriving at the solution of the original problem, which is often very hard to compute directly (Forster, 1995).

[2]To be more precise, the set of points satisfying the Karush-Kuhn-Tucker (KKT) necessary conditions for optimality along with the corresponding KKT multipliers form a manifold of dimension $m - 1$.

## 3 CONTINUATION METHOD

### 3.1 SOME BASICS ON MULTIOBJECTIVE OPTIMIZATION

A multiobjective optimization problem can be mathematically formalized as

$$
\min_{\theta \in \mathbb{R}^n} \begin{bmatrix} F_1(\theta) \\ \vdots \\ F_m(\theta) \end{bmatrix}, \tag{MOP}
$$

where $F_i : \mathbb{R}^n \to \mathbb{R} \; \forall \; i = 1, \ldots, m$ are the generally conflicting objective functions and $\theta$ the parameters we are optimizing over. In general, there does not exist a single point that minimizes all criteria simultaneously, the solution to (MOP) is the *Pareto set* of optimal compromises.

**Definition 1 (Miettinen (1998))** *A point $\theta^* \in \mathbb{R}^n$ is* Pareto optimal *if there does not exist another point $\theta \in \mathbb{R}^n$ such that $f_i(\theta) \leq f_i(\theta^*)$ for all $i = 1, \ldots, m$, and $f_j(\theta) < f_j(\theta^*)$ for at least one index $j$. The set of all Pareto optimal points is the* Pareto set, *which we denote by $P$. The set $F(P) \subset \mathbb{R}^m$ in the image space is called the* Pareto front.

The above definition is not very helpful for gradient-based algorithms. In this case, we need to rely on first order optimality conditions (also known as Karush-Kuhn-Tucker (KKT) conditions).

**Definition 2 (Hillermeier (2001))** *A point $\theta^* \in \mathbb{R}^n$ is called* Pareto critical *if there exist an $\alpha \in \mathbb{R}^m$ with $\alpha_i \geq 0$ for all $i = 1, \ldots, m$ and $\sum_{i=1}^m \alpha_i = 1$, satisfying*

$$
\sum_{i=1}^m \alpha_i \nabla F_i(\theta^*) = 0. \tag{1}
$$

In this work, we consider two objective functions, namely the empirical loss and the $\ell^1$ norm of the neural network weights. The Pareto set connecting the individual minima (at least locally), is also known as the regularization path. In terms of an MOP, we are looking for the Pareto set of

$$
\min_{\theta \in \mathbb{R}^n} \begin{bmatrix} \mathbb{E}_{(x,y) \sim \mathcal{D}}[\mathcal{L}(f(\theta, x), y)] \\ \frac{1}{n} \|\theta\|_1 \end{bmatrix}, \tag{MoDNN}
$$

where $(x, y) \in \mathcal{X} \times \mathcal{Y}$ is labeled data following a distribution $\mathcal{D}$, the function $f : \mathbb{R}^n \times \mathcal{X} \to \mathcal{Y}$ is a parameterized model and $\mathcal{L}(\cdot, \cdot)$ denotes a loss function. The second objective is the weighted $\ell^1$ norm $\frac{1}{n} \|\theta\|_1$ to ensure sparsity. Our goal is to solve MoDNN to obtain the regularization path. However, this problem is challenging, as the $\ell^1$ norm is not differentiable.

**Remark 3** *A common approach to solve the problem MOP (including all sorts of regularization problems) is the use of the* weighted sum method *using an additional hyperparameter $\lambda$:*

$$
\min_{\theta \in \mathbb{R}^n} F(\theta) = \sum_{i=1}^m \lambda_i F_i(\theta) \quad with \quad \lambda_i \geq 0 \; \forall i \in \{1, \ldots, m\} \quad and \quad \sum_{i=1}^m \lambda_i = 1. \tag{2}
$$

### 3.2 PROXIMAL GRADIENT METHOD

Given functions of the form $F_i = f_i + g_i$, such that $f_i : \mathbb{R}^n \to \mathbb{R}$ is convex and smooth, and $g_i$ is convex and non-smooth with computable proximal operator $\text{prox}_g$,

**Definition 4 (Proximal operator)** *Given a convex function $g : \mathbb{R}^n \to \mathbb{R}$, the proximal operator is*

$$
\text{prox}_g(\theta) = \arg\min_{\phi \in \mathbb{R}^n} \left\{ g(\phi) + \frac{1}{2} \|\phi - \theta\|_2^2 \right\}.
$$

In the problem MoDNN we have $F_1(\theta) = f_1(\theta) = \mathbb{E}_{(x,y) \sim \mathcal{D}}[\mathcal{L}(f(\theta, x), y)]$ and $F_2(\theta) = g_2(\theta) = \frac{1}{n} \|\theta\|_1$. The proximal operator $\text{prox}_{\frac{1}{n}\|\cdot\|_1}(\theta)$ has a simple closed form. This allows for an efficient implementation of Algorithm 1, which yields a single Pareto critical point for MOPs with objectives of such type. Tanabe, Fukuda, and Yamashita (2019) have proved that this algorithm converges to Pareto critical points.

---

**Algorithm 1** Multiobjective Proximal Gradient (Tanabe et al. (2019) )

---

**Input** Initialize $k = 0$.
**Parameter** $\theta^0 \in \mathbb{R}^n$, step size $h > 0$.
**Output**: $\theta^k$

1: **Compute** the descent direction $d^k$ by solving

$$d^k = \arg\min_{d \in \mathbb{R}^n} \left\{ \psi_{\theta^k}(d) + \frac{1}{2h}\|d\|_2^2 \right\}, \text{ where } \psi_\theta(d) = \max_{i=1,\ldots,m} \left\{ \nabla f_i(\theta)^T d + g_i(\theta + d) - g_i(\theta) \right\}$$

2: **if** $d^k = 0$, **STOP**
3: **Update** $\theta^{k+1} = \theta^k + d^k$
4: **Set** $k = k + 1$ and go to step 1

---

**Remark 5** *Proximal gradient iterations are "forward-backward" iterations, "forward" referring to the gradient step and "backward" referring to the proximal step. The proximal gradient method is designed for problems where the objective functions include a smooth and a nonsmooth component, which is suitable for optimization with a sparsity-promoting regularization term.*

### 3.3 A PREDICTOR-CORRECTOR METHOD

This section describes our continuation approach to compute the entire Pareto front of problem MoDNN. Fig. 1 shows an exemplary illustration of the Pareto front approximated by a finite set of points that are computed by consecutive predictor and corrector steps. After finding an initial point $\theta^0$ on the Pareto front, we proceed along the front. This is done by first performing a predictor step in a suitable direction. As this will take us away from the front (Hillermeier, 2001; Bieker et al., 2022), we need to perform a consecutive corrector step that takes us back to the front.

**Predictor step:** Depending on the direction we want to proceed in, we perform a predictor step simply by performing a gradient step or proximal point operator step:

$$\theta^{k+1} = \theta^k - \eta\nabla_\theta\mathbb{E}_{(x,y)\sim\mathcal{D}}[\mathcal{L}(f(\theta^k, x), y)], \tag{3}$$

$$\text{or} \quad \theta^{k+1} = \text{prox}_{\eta\|\cdot\|_1}(\theta^k). \tag{4}$$

Eq. 3 is the gradient step for the loss objective function (i.e., "move left" in Fig. 1) and 4 represents the shrinkage performed on the $\ell^1$ norm (i.e., "move down" in Fig. 1). Note that this is a deviation from the classical concept of continuation methods as described previously, where we compute the tangent space of a manifold. However, due to the non-smoothness, the Pareto set of our problem does not possess such a manifold structure, which means that we cannot rely on the tangent space. Nevertheless, this is not necessarily an issue, as the standard approach would require Hessian information, which is too expensive in high-dimensional problems anyway. The approach presented here is significantly more efficient, even though it may come at the cost that the predictor step is sub-

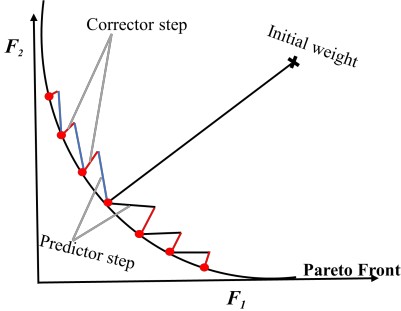

Figure 1: Sketch of the continuation method; The predictor steps are shown in black and blue for Eqs. 3 and 4 respectively. The corrector step is shown in red.

optimal. Despite this fact, we found in our numerical experiments that our predictor nevertheless leads to points close enough for the corrector to converge in a small number of iterations.

**Corrector step:** For the corrector step, we simply perform multiobjective gradient descent using the multiobjective proximal gradient method (MPG). As our goal is to converge to a point on the Pareto front, this step is identical for both predictor directions.

The method is summarized in Algorithm 2 for both directions, which only differ in terms of line 4.

---

**Algorithm 2** Predictor-Corrector Method

---

**Input**: Number of predictor-corrector runs $N_{cont}$, $P \leftarrow \{\}$
**Parameter**: Initial parameter $\theta^0 \in \mathbb{R}^n$, learning rate $\eta$
**Output**: $P$ approximate Pareto set

  1: **Compute** update to $\theta^0$ using *Algorithm 1* with initial value $\theta^0$ and step size $h = \eta$.
  2: **Update** $P = P \cup \{\theta^0\}$
  3: **for** $n = 1 : (N_{cont} - 1)$ **do**
  4:     **Compute** predictor $\theta_p^n$ by performing a predictor step from $\theta^{n-1}$ with Eq. 3 or 4
  5:     **Compute** corrector $\theta^n$ by applying *Algorithm 1* with initial value $\theta_p^n$ and step size $h = \eta$.
  6:     **Update** $P = P \cup \{\theta^n\}$
  7: **end for**

---

## 4   Numerical experiments

In this section, we present numerical experiments for our algorithm and the resulting improvements in comparison to the much more widely used weighted sum approach (Eq. 2).

In real-world problems, uncertainties and unknowns exist. Our previous approach of using the full gradient on every optimization step oftentimes makes it computationally expensive, as well as unable to handle uncertainties (Mitrevski et al., 2021). A stochastic approach is included in our algorithm to take care of these limitations. The stochasticity is applied by computing the gradient on mini-batches of the data.

### 4.1   Experimental settings

To evaluate our algorithm, we first perform experiments on the Iris dataset (Fisher, 1988). Although this dataset is by now quite outdated, it allows us to study the deterministic case in detail. We then extend our experiments to the well-known MNIST dataset (Deng, 2012) and the CIFAR10 dataset (Krizhevsky et al., 2014) using the stochastic gradient approach. The Iris dataset contains 150 instances, each of which has 4 features and belongs to one of 3 classes. The MNIST dataset contains 70000 images, each having 784 features and belonging to one of 10 classes. The CIFAR10 dataset contains 60000 images of size 32x32 pixels with 3 color channels and contains images belonging to one of 10 classes. We split the datasets into training and testing sets in a 80–20 ratio.

We use a dense linear neural network architecture with two hidden layers for both the Iris and MNIST datasets (4 neurons and 20 neurons per hidden layer, respectively), with ReLU activation functions for both layers. For the CIFAR10 dataset, two fully connected linear layers after two convolutional layers are used. Cross-Entropy is used as the loss function.

We compare the results from our algorithm against the weighted sum algorithm in Eq. 2, where $\lambda$ is the weighting parameter. We choose 44 different values for $\lambda$ for the Iris and MNIST dataset, and 20 different $\lambda$ values for CIFAR10, chosen equidistantly on the interval $[0, 1]$ and solve the resulting problems using the ADAM optimizer (Kingma & Ba, 2017).

The experiments for the Iris and MNIST datasets are carried out on a machine with 2.10 GHz 12th Gen Intel(R) Core(TM) i7-1260P CPU and 32 GB memory, using Python 3.8.8 while the CIFAR10 experiment is carried out on a compute cluster with an NVIDIA A100 GPU, 64 GB Ram and an 32-Core AMD CPU with 2.7GHz using Python 3.11.5. The source code is included as a zip file in the supplementary material (and will be made available via github after the review phase).

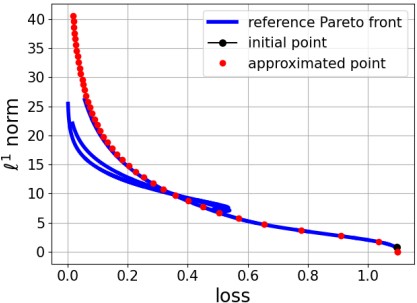

Figure 2: Pareto front approximation for the Iris dataset using *Algorithm 2* (red symbols) versus the reference Pareto front in "blue" (computed using the same algorithm with very small step sizes and many different initial conditions) with unscaled $\ell^1$ norm.

To illustrate the behavior of our algorithm, we first study the Iris dataset in a deterministic setting. To obtain a baseline, we have executed Algorithm 2 using very small step sizes. Interestingly, the Pareto set and front consist of multiple components, which we were only able to find by repeated application of the continuation method with random initial conditions (multi-start). The resulting solution is shown in blue in Fig. 2, where three connected components of the Pareto critical points are clearly visible. As this approach is much too expensive for a realistic learning setting (the calculations took close to a day for this simple problem), we compare this to a more realistic setting in terms of step sizes. The result is shown via the red symbols. Motivated by our initial statement on more sustainable network architectures, we have initialized our network with all weights being very close to zero (the black "●" in Fig. 2) and then proceed along the front towards a less sparse but more accurate architecture.[3]

As we do not need to compute every neural network parametrization from scratch, but use our predictor-corrector scheme, the calculation of each individual point on the front is much less time-consuming than classical DNN training. Moreover, computing the slope of the front from consecutive points allows for the online detection of relevant regions. Very small or very large values for the slope indicate that small changes in one objective lead to a large improvement in the other one, which is usually not of great interest in applications. Moreover, this can be used as an early stopping indicator to avoid overfitting, as we will see in the next example.

## 4.2 RESULTS

Motivated by these promising results, we now study the MNIST data set in a stochastic setting (i.e., using mini batches for the loss function). To obtain the initial point on the Pareto front using

---

[3]Due to symmetries, an initialization with all zeros poses problems in terms of which weights to activate, etc., see Bieker et al. (2022) for details.

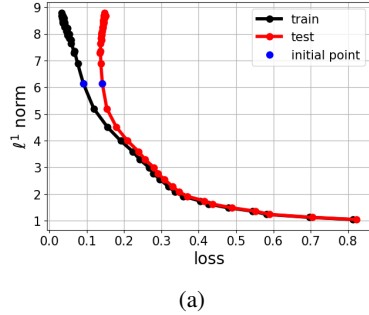

(a)

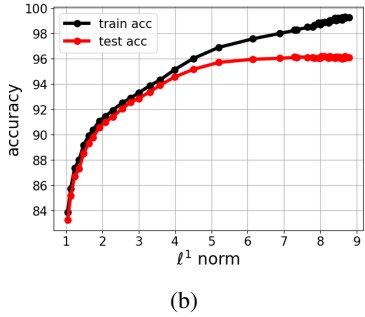

(b)

Figure 3: (a) The Pareto front for the MNIST data set (in black), where the initial point is shown in blue. The red curve shows the performance on the validation set. Non-sparse networks clearly tend to overfit. (b) The prediction accuracy versus $\ell^1$ norm, where the overfitting regime becomes apparent once more.

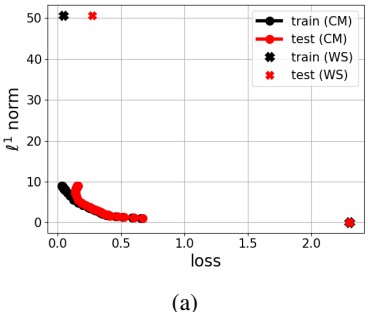 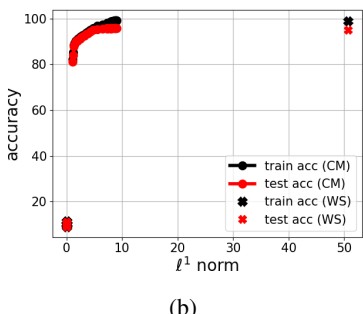

(a)                                             (b)

Figure 4: Comparison between the predictor-corrector (CM) approach and weighted sum (WS) approach the MNIST dataset. The figures show the same plots as Fig. 3 but include the WS solutions. A clustering around the sparse and non-regularized solutions is evident, even though we have used equidistantly distributed weights $\lambda$.

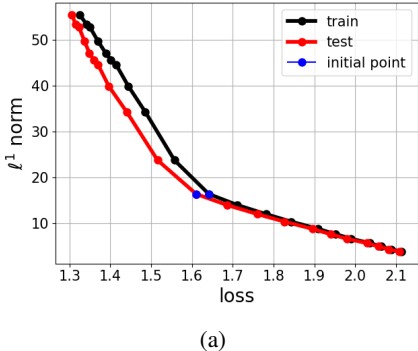 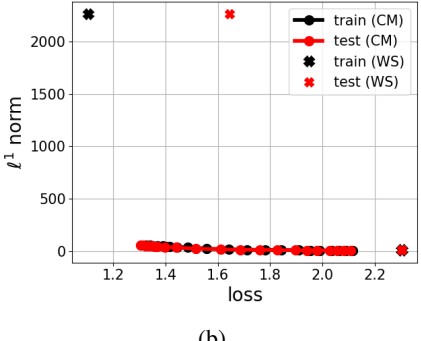

(a)                                             (b)

Figure 5: (a) The Pareto front for the CIFAR10 data set (in black), where the initial point is shown in blue. The red curve shows the performance on the test set. Non-sparse networks clearly tend to overfit. (b) The same plots as (a) but including the WS solutions. A clustering around the sparse and non-regularized solutions is evident in the WS method plot, even though we have used equidistantly distributed weights $\lambda$.

Algorithm 1, we perform 500 iterations. For the subsequent steps, 7 iterations are used for a predictor step and 20 for a corrector step repeated 43 times i.e., $N_{cont} = 44$. Figs. 3 (a) and (b) show the Pareto front and accuracy versus $\ell^1$ norm, respectively. In this setting, we have started with a point in the middle of the front in blue and then apply the continuation method twice (once in each direction). As indicated above, we observe overfitting for non-sparse architectures, which indicates that we do not necessarily have to pursue the regularization path until the end, but we can stop once the slope of the Pareto front becomes too steep. This provides an alternative training procedure for DNNs where in contrast to pruning we start sparse and then get less sparse only as long as we don't run into overfitting.

Fig. 4 shows the comparison of our results to the weighted sum (WS) approach in Eq. 2. For the WS approach, we compute 200 epochs per point on the front repeated 44 times i.e., $N_{\lambda} = 44$ equidistantly distributed weights $\lambda$, where $\lambda \in [0, 1]$. In total the WS approach needs 5 times as many epochs as the continuation method (*Algorithm 2*) and therefore 5 times as many forward and backward passes. We see also, that the WS approach results in clustering around the unregularized and very sparse solutions, respectively. In contrast, we obtain a similar performance with a much sparser architecture using continuation. This shows the superiority of the continuation method and also highlights the randomness associated with simple regularization approaches when not performing appropriate hyperparameter tuning.

To further test the performance of our algorithm on a higher-dimensional DNN, we extend our experiment to a much more complex neural network architecture with two convolutional layers and two linear layers having a total of $4,742,546$ parameters as shown in table *1*. We observe that our method provides well-spread points for the Pareto front. Fig. 5a shows the initial point obtained

| Data (parameters) | Method | Num iterations | Computation time (secs) | Accuracy | |
|---|---|---|---|---|---|
| | | | | Training Accuracy | Testing Accuracy |
| MNIST $(16, 330)$ | WS | $49.28e6$ | 1140 | **98.83**% | 94.94% |
| | CM | $9.3e6$ | 329 | 97.92% | **95.66**% |
| CIFAR10 $(4, 742, 546)$ | WS | $1.77e6$ | 575 | **68.81**% | 38.08% |
| | CM | $1.95e6$ | 1193 | 52.64% | **50.93**% |

Table 1: Settings and Results on MNIST and CIFAR10 dataset. For each metric, the best performance per architecture is in bold.

after 2000 iterations and the subsequent points obtained using the predictor step (7 iterations) and corrector step (25 iterations) for a total of 20 points on the front.

Fig. 5b shows the comparison of the CM with the WS for the CIFAR10 dataset and it can be observed once again—as for MNIST—that the WS yields clusters at the minima of the individual objectives and tends to overfit. For the WS method, an epoch size of 118 for each 20 points on the front is used with $\lambda \in [0, 1]$. The CM provides a good trade-off between sparsity and loss in DNNs, not just a very sparse DNN. Hence, we obtain a well-structured regularization path for nonlinear high dimensional DNNs.

Finally, we note that once an initial point on the Pareto front for the predictor-corrector method (CM) is identified, which is comparable to the complexity of the scalarized problem, the follow-up computations require fewer iterations and are much cheaper, when compared with the WS method.

## 5 CONCLUSION

We have presented an extension of regularization paths from linear models to high-dimensional nonlinear deep learning models. This was achieved by extending well-known continuation methods from multiobjective optimization to non-smooth problems and by introducing more efficient predictor and corrector steps. The resulting algorithm shows a performance that is suitable for high-dimensional learning problems.

Moreover, we have demonstrated that starting with sparse models can help to avoid overfitting and significantly reduce the size of neural network models. Due to the small training effort of consecutive points on the Pareto front, this presents an alternative, structured way to deep neural network training, which pursues the opposite direction than pruning methods do. Starting sparse, we increase the number of weights only as long as we do not obtain a too-steep Pareto front, as this suggests overfitting.

For future work, we will consider more objectives. Our approach works very well on extremely high-dimensional problems (e.g., CIFAR-10) with two objectives, where the Pareto front is a line. Extending our work to cases with more than two objective functions will require significant extensions to the concept of adaptive Pareto exploration (e.g., (Schütze et al., 2019). Since the Pareto set becomes a higher-dimensional object, it will no longer be useful to compute the entire set but steer along desired directions in order to meet a decision maker's desired trade-off.

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
