# OpenReview forum: "A multiobjective continuation method to compute the regularization path of deep neural networks"
_ICLR.cc/2024/Conference — Submitted to ICLR 2024_

### Official Review · Reviewer_dHK2 · 2023-10-30

**Soundness:** 2 fair
**Presentation:** 2 fair
**Contribution:** 1 poor
**Rating:** 3
**Confidence:** 4

**Summary:**

This paper studies multi-objective optimization problem with $\ell_1$ regularization. The paper claims that they extend the target problem from linear to high-dimensional non-linear problems. Numerical experiments on MNIST demonstrate the efficacy of the proposed optimization schema.

**Strengths:**

- The paper is written well and easy to follow.
- The target problem is important to the community.

**Weaknesses:**

- Missing references.
 The authors claim in introduction that `Very recently, there was a first attempt to extend the concept of regularization paths
 to DNNs by means of treating the empirical loss and sparsity.`
However, dating back to 2020, the problem has been studied in OBProxSG where the multi-objective could be considered as a weighted sum of each individual target objective.

   [1] Orthant Based Proximal Stochastic Gradient Method for $\ell_1$-Regularized Optimization, 2020.

- Experiments are not sufficient. I am concerned about the experiments. In particular, the paper studies multi-objective problems. However, in the experiment, only cross-entropy is considered as loss function. Where are other losses to form a real multi-objective problem? Meanwhile, the network and dataset are too simple to validate the efficacy.

**Questions:**

See the weakness.

---

> ### Author Response · Authors · 2023-11-22
>
> Thank you for your detailed and thoughtful review of our paper.  Regarding your comments, suggestions and questions, we have carefully considered them, and below are answers to your questions. In summary, we have;
>
> * Added a second, much more complex example using the CIFAR10 dataset, and demonstrated that the advantages of our approach also hold in much higher dimensions.
>
> * Argued why our method is applicable to more complex examples than many alternative algorithms.
>
> * Argued why our method tends to be more robust than standard training approaches.
>
> * Increased the discussion on numerical details of our approaches and the numerical experiments.
>
> * Added the references suggested by the reviewer to the related work section of our paper.
>
> We hope that this convinces the reviewer that our work is indeed novel, as it is the first method that allows for computing the entire front for non-convex problems of such large dimensions.
>
> **All changes in the updated PDF version of our paper have been marked in blue.**
>
> **Questions and Answers:**
>
> 1. Missing references: The authors claim in introduction that Very recently, there was a first attempt to extend the concept of regularization paths to DNNs by means of treating the empirical loss and sparsity. However, dating back to 2020, the problem has been studied in OBProxSG where the multi-objective could be considered as a weighted sum of each individual target objective. [1] Orthant Based Proximal Stochastic Gradient Method for -Regularized Optimization, 2020.
>
> Answer: **You are correct that the mentioned paper goes in a similar direction, and we have added it to our discussion on related work. However, similar to other works, this paper does not address the computation of the entire Pareto front. In this regard, our paper in fact is the very first that truly solves a multiobjective optimization problem in such high dimensions, both in the machine learning and the optimization literature.**
>
> 2. Experiments are not sufficient:  I am concerned about the experiments. In particular, the paper studies multi-objective problems. However, in the experiment, only cross-entropy is considered as loss function. Where are other losses to form a real multi-objective problem? Meanwhile, the network and dataset are too simple to validate the efficacy.
>
> Answer: **A second, much more complex experiment using CIFAR10 and a more complex model has been included in the updated version of our paper. In terms of the problem not being multi-objective, we disagree in the sense that the problem truly does have a Pareto front for the two tasks we consider. On the other hand, we agree that multiple main tasks are just as interesting, in particular when this results in more than two objectives. We have added a discussion on this aspect in the outlook of the paper. As the goal of this work is to provide the Pareto front for such a large-dimensional problem for the first time, we would rather leave this additional challenge for future work, and then also address the question of steering on higher-dimensional sets.**

---

### Official Review · Reviewer_4ykt · 2023-10-30

**Soundness:** 3 good
**Presentation:** 3 good
**Contribution:** 3 good
**Rating:** 6
**Confidence:** 5

**Summary:**

The paper presents a method to travel on the Pareto front for a two-objective optimization problem and shows how to apply it to compute the regularization path for deep NNs using the L1 penalty.

**Strengths:**

An interesting approach that seems to work well in traveling on the Pareto front.

**Weaknesses:**

- The title and the paper over-claims to work on multi-objective problems, but actually the method is only for two objective problems. For more than two objectives, it seems to me that the method will not explore the entire Pareto front.
- A comparison is missing with the standard approach to obtain the regularization path that starts with no penalty and gradually increases the L1 penalty lambda and optimize the penalized loss for each value of lambda.
- The L1 penalty is known to bias the weights and prevent the model from fitting the data well. In that respect non-convex penalties such as SCAD/MSC or the L0 penalty are preferred.

**Questions:**

- Can the proposed method work on more than two objective functions?
- How does the method compare din computation time and accuracy with the standard approach for obtaining the regularization path described above?
- Can the proposed method work with the SCAD/MCP penalties?

---

> ### Author Response · Authors · 2023-11-22
>
> Thank you for your detailed and thoughtful review of our paper.  Regarding your comments, suggestions, and questions, we have carefully considered them, and below are answers to your questions. In summary, we
>
> * added a second, much more complex example using the CIFAR10 dataset, and demonstrated that the advantages of our approach also hold in much higher dimensions
> * argue why our method is applicable to more complex examples than many alternative algorithms.
> * argue why our method tends to be more robust than standard training approaches.
> * increased the discussion on numerical details of our approaches and the numerical experiments.
> * added some references to related work.
>
> We are convinced that this is the first work that allows for computing the entire front for non-convex problems of such large dimensions.
>
> **All changes in the updated PDF version of our paper have been marked in blue.**
>
> **Questions and Answers:**
>
> 1. Can the proposed method work on more than two objective functions?
>
> Answer: **We are convinced that our proposed method works on more than two objective functions. From a computational perspective, both the predictor and corrector steps should not be any more expensive. However, given that in more than two objective functions, the Pareto front is no longer a line, the task of selecting a predictor direction becomes more challenging. We have added some references and a short discussion to address this concern. However, the task of our paper was to compute the entire front of very high-dimensional, non-convex fronts for the first time, such that we would like to leave the explicit treatment of more objectives for future work. We now say so in the conclusion.**
>
> 2. How does the method compare din computation time and accuracy with the standard approach for obtaining the regularization path described above?
>
> Answer: **The comparison of our method with standard approach as you have described can be seen in the comparison of our method with weighted sum where the weighting parameter lambda takes the values 1 for sparsity only $\ell^1$ norm, and lambda = 0 loss only. The results of the comparisons have been included as a table in the uploaded PDF version of our paper.**
>
> 3.	Can the proposed method work with the SCAD/MCP penalties?
>
> Answer: **Thank you for bringing this up. We believe that including other penalties such as the SCAD (Smoothly Clipped Absolute Deviation) and MCP (Minimax Concave Penalty) which have non-convex shapes similar to the $\ell^1$ norm as conflicting objectives to the loss function would be possible for our method. This could be the case when exploring more than two objective functions.**

---

### Official Review · Reviewer_PWoo · 2023-11-03

**Soundness:** 3 good
**Presentation:** 3 good
**Contribution:** 3 good
**Rating:** 3
**Confidence:** 4

**Summary:**

This paper proposes to extend the concept of regularization path from linear model to deep neural networks. In detail, the authors formulate the problem as a multi-objective composed of empirical loss as well as sparsity regularization. The authors propose an efficient approximation algorithm to recover the entire Pareto front as the regularization path. The proposed method is composed of two parts, the first part is a stochastic gradient descent combined with a proximal mapping. The other step is a multi-objective update step, which maps the solution after the gradient step back to the required Pareto set. To validate the proposed method, several experiments are conducted to validate the efficacy. In specific, the authors conduct experiments on MNIST a widely used dataset and validate the empirical Pareto front.

**Strengths:**

This paper proposes a novel view to solve the sparsity constrained problem in deep learning. The idea of using multi-objective to formulate the problem is novel. Additionally, an efficient predictor and corrector algorithms is proposed to find the Pareto front of the training process. The authors also put emphasis on the regularization path which can give a deeper understanding of the training process. The method is also validated with a widely used dataset.

**Weaknesses:**

For the topic of extending regularization path from linear model to deep learning, several references[1][2][3] are missed.For [1], it utilizes Bregman Iteration to explore sparsity when training deep neural network where regularization can be generated during the iteration.  For [2], it proposed an efficient way to generate the reguarlization path from simple to complex via exploring inverse scale space.  For [3], it extends lasso from linear model to deep neural networks, regularization path is also discussed in this paper.

In addition, could the authors further explain the benefit of finding such Pareto front during training neural networks.

For the Multiobjective Proximal Gradient algorithm, for the setting training with only cross entropy loss, it seems that the formulation is similar to the Bregman Iteration as discussed in [4], could the authors give a further discussion?

For the experiment parts, the current experiments are not sufficient. Only showing the performance on MNIST is not persuasive. It would be better for the authors to add experiments on more datasets.

Furthermore, could the authors illustrate whether the proposed method could be used to find the Pareto front of a series of Lottery Ticket Subnetworks[5].

[1] Bungert, Leon, et al. "A Bregman learning framework for sparse neural networks." The Journal of Machine Learning Research 23.1 (2022): 8673-8715.

[2]Fu, Yanwei, et al. "Exploring structural sparsity of deep networks via inverse scale spaces." IEEE Transactions on Pattern Analysis and Machine Intelligence 45.2 (2022): 1749-1765.

[3] Lemhadri, Ismael, Feng Ruan, and Rob Tibshirani. "Lassonet: Neural networks with feature sparsity." International conference on artificial intelligence and statistics. PMLR, 2021.

[4] Osher, Stanley, et al. "An iterative regularization method for total variation-based image restoration." Multiscale Modeling & Simulation 4.2 (2005): 460-489.

[5]Frankle, Jonathan, and Michael Carbin. "The lottery ticket hypothesis: Finding sparse, trainable neural networks." arXiv preprint arXiv:1803.03635 (2018).

**Questions:**

Please refer to weakness.

---

> ### Author Response · Authors · 2023-11-22
>
> Thank you for your detailed and thoughtful review of our paper.  Regarding your comments, suggestions and questions, we have carefully considered them. To address the mentioned weaknesses, we
> * Added a second, much more complex example using the CIFAR10 dataset, and a CNN architecture containing two convolutional layers and two linear layers with a total number of 4,742,546 parameters. We demonstrate that the advantages of our approach also hold in much higher dimensions.
> * Argue why our method is applicable to more complex examples than many alternative algorithms.
>
> * Argue why our method tends to be more robust than standard training approaches.
>
> * Included the discussion on numerical details of our approaches and the numerical experiments.
>
> * Added the references you mentioned to the related work section of our paper.
>
> We believe that our method shows some advantages over the mentioned references. The mentioned references all explore the goal of inducing sparsity into DNNs. In contrast, the goal of our approach is not only to enforce sparsity but also to ensure a good trade-off between sparsity and loss, hence the concept of multiobjective optimization and the goal to identify the entire Pareto front. With our predictor-corrector (CM) method, this Pareto set of optimal compromises between sparsity and loss is obtained giving a decision maker series of points to choose from at the same time understanding the tradeoff between both objectives. We are convinced that this is the first work that allows for computing the entire front for non-convex problems of such large dimensions.
>
> **All changes in the updated PDF version of our paper have been marked in blue.**

---

> > ### Comment · Reviewer_PWoo · 2023-11-23
> >
> > I appreciate the response, but I will keep my score.

---

### Official Review · Reviewer_oW7r · 2023-11-05

**Soundness:** 3 good
**Presentation:** 3 good
**Contribution:** 3 good
**Rating:** 6
**Confidence:** 3

**Summary:**

This paper proposes a multiobjective optimization method for non-smooth problem with efficient predictor and corrector step by approximation of Pareto front. It can extend the regularization paths from linear models to nonlinear high-dimensional deep learning models. The method is used to train neural network starting sparse, which can help avoid overfitting by early stop.

This is the comments from the fast reviewer.

**Strengths:**

Strength:
1. The idea using approximation of Pareto front to improve multiobjective optimization on non-smooth problem to extend regularization path to high-dimensional nonlinear deep learning model is interesting and effective.

2. The results shows the method can help avoid the overfitting problem and obtain models with low loss and sparsity.

3: The paper provides thorough related background information in Section 3, making it easy to follow.

4: The predictor-corrector scheme is innovative, avoiding clustering around the unregularized and very sparse solutions, in comparison with weighted sum approach.

**Weaknesses:**

1: You highlight the efficiency of your method in several aspects, such as reducing the computational expense by avoiding the computation of the full gradient in Section 4, and also provide your experimental settings. While your experimental settings are detailed, the inclusion of metrics on computational time would significantly enhance the evaluation of your method's efficiency.

2: You claim that the algorithm shows a performance that is suitable for high-dimensional learning problems, may should be supported by more results extended to deeper DNNs including non-linear layers, and more complex datasets just as you mention in Section 5.

3: Have you compared the performance of your method with existing regularization path generation methods?

4: One of your key merits is avoiding overfitting, and it would strengthen the claim to conduct additional experiments.

**Questions:**

1: You highlight the efficiency of your method in several aspects, such as reducing the computational expense by avoiding the computation of the full gradient in Section 4, and also provide your experimental settings. While your experimental settings are detailed, the inclusion of metrics on computational time would significantly enhance the evaluation of your method's efficiency.

2: You claim that the algorithm shows a performance that is suitable for high-dimensional learning problems, may should be supported by more results extended to deeper DNNs including non-linear layers, and more complex datasets just as you mention in Section 5.

3: Have you compared the performance of your method with existing regularization path generation methods?

4: One of your key merits is avoiding overfitting, and it would strengthen the claim to conduct additional experiments.

5. It seems that finding the initial point and using randomly multi-start to find components of Pareto front cost a lot. With more complex network structure, it is possible that finding the initial point be a bottleneck which is hard to accelerate.

6. With stochastic gradient, it is possible that the initial points are different between experiments. Will different initial points affect the performance or the iteration times of the algorithm?

7. What is the method to decide when should the training stop before when the slope of the Pareto front becomes too steep. In practical settings, it seems to complex and tricky to achieve the early stopping.
8. One missing related reference: DessiLBI: Exploring Structural Sparsity on Deep Network via Differential Inclusion Paths. ICML2020

---

> ### Author Response · Authors · 2023-11-22
>
> Thank you for your detailed and thoughtful review of our paper.  Regarding your comments, suggestions, and questions, we have carefully considered them, and below are answers to your questions. In summary, we have;
> * Added a second, much more complex example using the CIFAR10 dataset, and demonstrated that the advantages of our approach also hold in much higher dimensions.
> * Argue why our method is applicable to more complex examples than many alternative algorithms.
> * Argue why our method tends to be more robust than standard training approaches.
> * Increased the discussion on the numerical details of our approaches and the numerical experiments.
> * Added some references to related work.
> We are convinced that this is the first work that allows for computing the entire front for non-convex problems of such large dimensions.
>
> **All changes in the updated PDF version of our paper have been marked in blue.**
>
> **All the answers below are numbered according to the question number asked.**
>
> ## ANSWERS
> 1. We have included a table detailing the metrics and experimental settings for our continuation method (CM) and weighted sum (WS) in the updated PDF version of our paper.
>
> 2.    We have implemented both our continuation method (CM) and the weighted sum (WS) method on CIFAR10 using a CNN model architecture containing two convolutional layers and two fully connected linear layers with a total number of 4,742,546 parameters. We demonstrate that our CM still generalizes better than the WS and provides a set of well distributed points on the front. Updated results of our new experiments can be seen in the updated PDF version of our paper and we will be adding an appendix section in the final paper submission.
>
> 3. Yes, we implemented methods such as the early stopping and grid search (tuning of different hyperparameter values learning rate, weight decay). We observed that for both CM and WS, these methods tend to have similar effects. E.g., early stopping reduces the accuracy of both methods, and different values of the learning rate and weight decay tend to affect the rate of convergence to the Pareto front for both methods. We will be including a detailed discussion on these in the appendix section which will be added in the final submission.
>
> 4. By having added the CIFAR10 example, we believe that we can now make a much stronger case for our claim that we can avoid overfitting.
>
> 5. Finding the initial point is approximately as expensive as training a neural network (NN) using a standard single-objective method, probably a little cheaper as we do not need to thrive for the lowest possible loss. However, the consecutive predictor and corrector steps remain very cheap even for very large architectures such as our CIFAR10 example, such that the benefit of computing the front comes at little extra cost.
>
> 6. We have made extensive tests regarding multiple runs. Since our method does not aim for the smallest loss when finding the initial point on the front (cf. our response to question 5), we observe only very small variations in terms of repeated experiments. We will add more details to this in the appendix section of the final paper which will be added in the final submission.
>
> 7. The reviewer raises a good point here. The focus of our paper was first to demonstrate that we can obtain the entire Pareto front even for extremely high-dimensional problems. The question of early stopping is of vital importance, though, and we will add a brief discussion in the appendix section of the final paper submission, where we argue that a small validation set should suffice to detect the turn-around point in terms of the out-of-sample loss.
>
> 8. Thank you for the reference. We have added it to our paper and included it in the discussion on related work.

---

### Author Response · Authors · 2023-11-22
**General response statement**

We would like to express our gratitude for the detailed and very helpful reviews. We have addressed all the reviewers’ comments and are confident that this has significantly improved the quality of our paper. In particular, we have;
 * Added CIFAR10 as a much more complex example and trained a CNN of much higher dimension using our algorithm. A detailed comparison to the weighted sum method demonstrates the advantages a true multiobjective treatment provides, most importantly in terms of trade-off selection and avoiding overfitting.

* Added several references to related work that were suggested by some of the reviewers, as well as included a discussion regarding where we see the benefits of our work.

In summary, we are convinced that our paper provides a valuable addition to the Machine Learning literature, and we are convinced that this is the first work that allows for computing the entire front for non-convex problems of such large dimensions.

All changes in the updated PDF version of our paper have been marked in blue.

---

### Meta-Review · Area_Chair_CA7T · 2023-12-09

**Metareview:**

This paper extends the concept of regularization paths from linear models to deep neural networks (DNNs) by formulating a multi-objective problem that combines empirical loss and sparsity regularization. The authors propose an efficient approximation algorithm to recover the entire Pareto front, which consists of a stochastic gradient descent with a proximal mapping and a multi-objective update step. Experiments on the MNIST dataset are used to validate the approach.

Strengths:
The paper presents an innovative approach to address the sparsity constrained problem in DNNs using multi-objective optimization.
The authors propose a predictor-corrector algorithm to find the regularization path, providing insight into the training process.
The method can potentially help avoid overfitting and obtain models with a good balance of low loss and sparsity.
The background information is thoroughly presented, making the paper accessible.
The multi-objective scheme avoids clustering at the extremities of the regularization path, which is a notable improvement over the weighted sum approach.

Weaknesses:
1) The paper may overstate its applicability to multi-objective problems since it appears primarily focused on two-objective scenarios.
A comparison with standard approaches to obtain regularization paths, such as varying the L1 penalty lambda, is missing.
2) The L1 penalty can introduce bias in the weights, and alternatives like SCAD/MSC or L0 penalties may be preferable but are not discussed.
3) The literature review misses several relevant references that have explored similar concepts in extending regularization paths to DNNs.
4) The benefits of finding the Pareto front during neural network training could be more explicitly discussed.
5) The experiment section is limited, with only the MNIST dataset explored, raising questions about the method's performance on more complex datasets and deeper networks.

While the proposed approach is innovative and has potential advantages, there are concerns about the method's generalizability and the lack of comparison with existing methods. More extensive experiments and a broader discussion on the applicability and benefits of the approach would strengthen the current version of the paper.

**Justification For Why Not Higher Score:**

After the rebuttal period, two reviewers suggest rejection, one reviewer suggests accept, while one weakly accepts. So the final decision follows majority.

**Justification For Why Not Lower Score:**

N.A.

---

### Decision · Program_Chairs · 2024-01-16

Reject